# Assessment of the Soil Erosion Response to Land Use and Slope in the Loess Plateau—A Case Study of Jiuyuangou

**Chenlu Huang [1], Qinke Yang [1,*], Xiayu Cao [2] and Yuru Li [3]**

[1] Department of Urban and Environment Sciences, Northwest University, Xi'an 710127, China; nwuhcl@163.com

[2] Upper and Middle Yellow River Bureau, Yellow River Conservancy Commission, Xi'an 710127, China; xiacyuc@163.com

[3] Shaanxi Surveying and Mapping Bureau, Xi'an 710127, China; liyuru@stumail.nwu.edu.cn

* Correspondence: qkyang@nwu.edu.cn; Tel.: +86-13609259298

**Abstract:** Soil erosion is a serious environmental problem in the Loess Plateau, China. Therefore, it is important to understand and evaluate soil erosion process in a watershed. In this study, the Chinese Soil Loss Equation (CSLE) is developed to evaluate the soil loss and analyze the impact of land use and slope on soil erosion in Jiuyuangou (JYG) watershed located in the hilly-gullied loess region of China 1970–2015. The results show that the quantities of soil erosion decreased clearly from 1977 to 2015 in the study area, which from 2011 (t/km$^2$·a) in 1977 to 164 (t/km$^2$·a) in 2004 and increased slowly to 320 (t/km$^2$·a) in 2015. No significant soil erosion (<300 t/km$^2$·a) changed in JYG watershed, which increased dramatically from 8.93% to 69.34% during 1977–2015. The area of farmland in this study area has been reduced drastically. Noting that the annual average soil erosion modulus of grassland was also showing a dropped trend from 1977 to 2015. In addition, the study shows that the annual average soil erosion modulus varied with slope gradient and the severe soil erosion often existed in the slope zone above 25°, which accounted for 4657 (t/km$^2$·a) in 1977 and 382.27 (t/km$^2$·a) in 2015. Meanwhile, soil erosion of different land-use types presented the similar changing trend (declined noticeably and then increased slowly) with the change of slope gradient from 1977 to 2015. Combined the investigations of extreme rainfall on 26 July 2015 for JYG watershed, the study provides the scientific support for the implementation of soil and water conservation measures to reduce the soil erosion and simplify Yellow River management procedures.

**Keywords:** soil erosion; CSLE; land use; slope gradient; Jiuyuangou watershed

## 1. Introduction

Soil erosion is one of the most severely environmental problems in the world [1,2], as it directly causes soil and water losses and soil degradation [3,4]. The Chinese government has implemented a lot of measures in the Loess Plateau to improve and conserve soil and water conservation measures since the middle of the twentieth century [5]. As a crucial research project, many researchers analyzed the interaction effects of important influencing factors (land use, vegetation cover and topography) on the soil erosion process [6,7].

There are various models that have been developed to assess the soil loss by water such as the WEPP—Water Erosion Prediction Project [8], USLE—Universal Soil Loss Equation [9], RUSLE—the Revised Universal Soil Loss Equation [10] and EPIC—the Erosion Productivity Impact Calculator [11]. Those models were all widely applied in many countries to evaluate the soil erosion especially the USLE, which evaluates the soil loss better with modifications according to its own condition. However,

USLE developed according to the sheet and rill erosion to estimate long-term average annual soil loss [9]. Consequently, the results that was calculated through this method often over-estimate slight erosion and under-estimate severe erosion [12]. Therefore, the Chinese Soil Loss Equation (CSLE) was developed to evaluating the average annual soil loss by water based on the specific characteristics in Loess Plateau, China [13]. This model is more suitable for the quantitative assessment of soil loss by water in the area mainly influenced by a great number of soil and water conservation measures (engineering measures, biology measures and tillage measures) and steeper slope steepness in China [14]. The fourth national census was launched for soil erosion evaluation in China from 2010 to 2012 by using the CSLE combined with a field survey, and achieved the goals of investigating the total area of soil loss [15]. Some researchers used the CSLE model to evaluate soil erosion and found that the CSLE was more effective than other commonly used methods in Wuqi county of Shaanxi Province, China [16,17]. Additionally, combining with the geographic information system (GIS) and remote sensing technology, this study aids to calculate and explain the process of soil erosion and analyzes the interaction between the land use, vegetation cover, topographic and soil erosion.

Substantial efforts have been made on the development of soil erosion models [18,19]. Land use, vegetation cover and topography are all key factors, which have a substantial impact on erosion processes [20–22]. Liu et al. further compared and analyzed the change of land use and soil and water conservation measures (forest, grass, dam and terrain) over the Yellow River Basin, and found that the land use in recent years improved by 30% compared with that in the 1970s and the sediment load decreased in recent years [15]. Sun et al. took the Loess Plateau as the research area to analyze the soil erosion under the different types of land use, vegetation cover and topography. The study found that the growing vegetation cover slows down the soil erosion rate from 2000 to 2010 [6]. However, the soil erosion remains unclear in some small typical watersheds and it is also ambiguous how different land uses/covers and topographies interact with soil erosion in the long time serial.

Therefore, it is urgently needed to combine the quantification assessment of soil loss with the long-term survey data to assess the intensity of soil erosion and to analyze the interaction effects of land uses/covers, topography on soil erosion in the small watersheds. So, in this study, we chose the Jiuyuangou (JYG) watershed as a study area, which was located in northern Shaanxi, China, one of the representative watersheds in the hilly loess region, where a field research station of soil and water conservation was built in 1952 on the Loess Plateau. With the effective soil and water conservation and the improved vegetation situation in this typical watershed, the soil erosion risk relieved noticeably. The primary objective of this study is as follows. (i) Evaluating the soil erosion and its impact factors in the JYG watershed from 1977 to 2015. (ii) Estimating the influence of land use, vegetative cover and topography on the soil erosion of the JYG watershed. (iii) Combining results of this paper with soil erosion after the 26 July in 2017 extreme rainstorm to provide some information on the application of CSLE and implementation of soil and water conservation measures in other watersheds.

## 2. Materials and Methods

### 2.1. Study Area

The Jiuyuangou (JYG) watershed (110°16′ E–110°26′ E, 37°33′ N–37°38′ N) located in the left bank of Wuding River Basin, northern Shaanxi of China (Figure 1). The region was known as the loess hilly and gully area of the Loess Plateau with the gully densities of 5.34 km/km$^2$. It covers an area of 70.7 km$^2$, with altitude between 820 and 1180 m above sea level. The climate is semi-humid, with an average annual temperature of 8 °C (Celsius) and the average annual evaporation is 1519 mm. The average precipitation is 475.1 mm and there is a high-intensive precipitation from June to September indicating significant inter-annual variability, the precipitation in 1964 was more than 735.3 mm are three times than that in 1956 (with just 232 mm). The average annual runoff is 0.275 km$^3$ and the runoff depth is 39.2 mm. The average sediment load is 0.59 million t (ton), and the largest sediment load is 9.59 million t in 1977, the smallest value occurred in 1980–1990. Loess soil, as the main soil type of the

basin with good permeability and high erosion, usually located on the bedrock or red clay and is the main skeleton of the watershed. Vegetation mainly composes of forest patches (*Robinia pseudoacacia* and *Pinus tabulaeformis*), shrub *(Caragana microphylla* and *Amorpha fruticosa*), grass *(Medicago sativa* and *Astragalus adsurgens*) and so on. Restoration in this area started in the 1980s, with nearly all farmland on the slopes being restored by anthropogenic activities or abandoned for natural recovery.

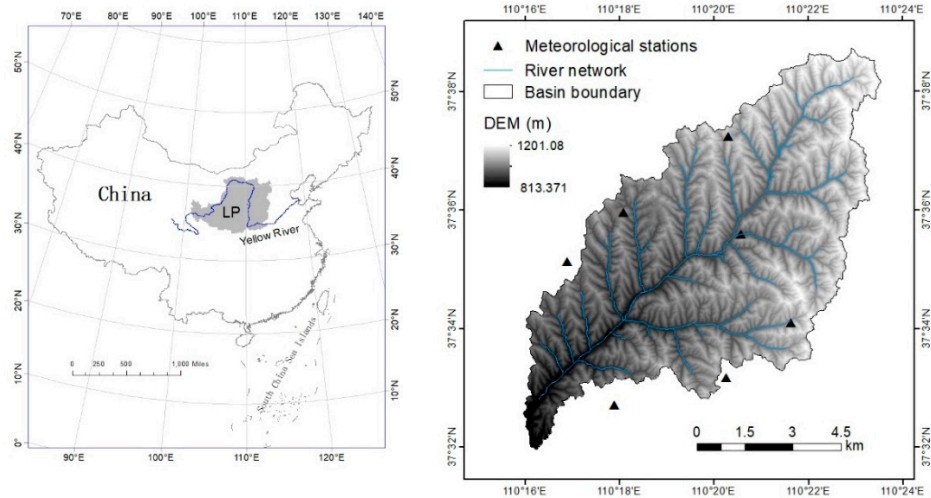

**Figure 1.** The locations of the Jiuyuangou watershed and LP (Loess Plateau) in China.

## 2.2. Materials

The satellite image, rainfall, soil, soil and water conservation measures data and the Digital Elevation Model (DEM) were used to estimate soil erosion in the JYG watershed. The relevant details about above information are presented in Table 1. The satellite image including an aerial scanning image in 1977, Google image map in 2004 and high-resolution map in 2015 as an assist map to interpret land uses; Landsat MSS image in 1975, Landsat TM image in 2004 and 2015 were used to estimate the vegetation covers in JYG watershed derived from GEE. Daily precipitation data in 1977, 2004 and 2015 were used to calculate the rainfall erodibility. The soil data was used as the basic data for calculating the soil erodibility factor, which derived from the accumulation of previous data. Soil and water conservation measures data were mainly based on surveys and statistics previous investigated in the basin. DEM, interpolated from contour maps, was used to calculate the LS factors then derived the tillage and gully erosion factors.

**Table 1.** Data preparation in this study.

| Data Type | Source | Resolution/Scale | Using |
|---|---|---|---|
| Aerial scanning image in 1977 | Provide by the "Rainstorm erosion project" | 1 m | Calculate factor B, T |
| Google image map in 2004 | Download from "91weitu" | 0.5 m | |
| High-resolution map in 2015 | Provide by the "Erosional gully monitoring project" | about 1.79 m | |
| Landsat MSS image in 1975 | Available on the Google Earth Engine | 80 m | |
| Landsat TM image in 2004, 2015 | | 30 m | |
| Precipitation | Meteorological Bureau | Table | Calculate factor R |
| Soil | From previous research | 5 m | Calculate factor K |
| Soil and water conservation measures statistics | Investigation statistics | based basin units | Calculate factor E |
| Digital elevation model (DEM) | Interpolated by contour | 5 m | Calculate factors LS, g |

*2.3. Estimating Soil Erosion using the CSLE Model*

This study used the Chinese Soil Loss Equation (CSLE), which is a soil loss model that considers the relationship between the soil erosion and its impact factors (rainfall erosivity, soil erodibility, slope steepness, slope length, vegetation covers, biological, engineering and tillage control measures) [13] to analyze the soil loss and the factors that influence it. Different from USLE, making itself being able to be applied to China, CSLE considering the soil loss with biological-control, engineering-control and tillage-control measures to replace the soil and water conservation measures in USLE and adding the steep slope factor based on measured data in the Loess Plateau [15]. CSLE played a good role in the investigation of fourth national soil erosion conditions [23] from 2010 to 2012 and soil erosion survey plot in Shanxi province [17]. The equation was as follows:

$$A = R \times K \times L \times S \times B \times E \times T \tag{1}$$

In Equation (1), *A* is the annual average soil erosion amount (t/(ha·a)), 1 t/(ha·a) = 100 t/(km$^2$·a); *R* is the factor of rainfall erosivity (MJ·mm/(ha·h·a)); *K* is the factor of soil erodibility (t·ha·h/(ha·MJ·mm)); L, S, B, E and T are the dimensionless factor of slope length, slope steepness, biomass-control, engineering-control and tillage practices in water and soil conservation respectively. The biomass-control practices (B factor) in CSLE similar with the cover and management measures (C factor) in USLE, which can reflect the land uses impact on soil erosion. Engineering-control practices (E factor) are the measures taken by reducing or preventing the soil erosion through change slope to terrace and construct check-dams. Tillage practices (T factor) refer to the measures to achieve the effect of water, soil and fertilizer conservation during the cultivation process, which mainly include the contouring and horizontal ditch.

2.3.1. Rainfall Erosivity Factor (R)

The rainfall erosivity factor is an important driving force of the rain affecting soil erosion. Wischmeier and Smith (1958) [24] calculated the rainfall erosivity through multiplying by the rainfall kinetic energy and maximum 30-minute rain intensity and applied it to the USLE to predict the soil erosion. The daily rainfall data is chosen in this study to estimate the annual rainfall erosivity according to the method of Zhang et al. (2002) [25], which is widely used in China [26–28]. The calculating method of annual rainfall erosivity is as follows:

$$M_i = \alpha \sum_{j}^{k} \left( D_j \right)^{\beta} \tag{2}$$

$$\beta = 0.8363 + 18.144 \cdot P_{d_{12}}^{-1} + 24.455 \cdot P_{y_{12}}^{-1} \tag{3}$$

$$\alpha = 21.586 \cdot \beta^{-7.1891} \tag{4}$$

*Mi* is the half-month rainfall erosivity (MJ·mm·hm$^{-2}$·h$^{-1}$), and *K* is the number days in the half-month, *Dj* is the effective rainfall for day *j* in one half-month, which is equal to the actual rainfall if its value is greater than the threshold value (12 mm, the standard value in China's erosive rainfall), otherwise, *Dj* is equal to zero. The terms of $\alpha$ and $\beta$ are the model parameters. $P_{d12}$ and $P_{y12}$ are the daily average more than 12 mm and yearly average rainfall that is for days with rainfall more than 12 mm respectively. The IDW (Inverse Distance Weight) method is used to interpolate the rainfall erosivity.

2.3.2. Soil Erodibility Factor (K)

The soil erodibility factor is the soil-loss rate per unit of erosion index for a specified soil in the standard plot condition [10], which is the primary indicator for assessing soil loss and related to the soil

properties including soil texture, organic matter and so on [29]. In this study, the K factor is calculated by the following equation:

$$K = \{0.2 + 0.3 \, exp[-0.0256 \, SAN(1 - SIL/100)]\} \times \left(\frac{SIL}{CLA+SIL}\right)^{0.3}$$
$$\times \left[1.0 - \frac{0.25C}{C+exp(3.72-2.95C)}\right]$$
$$\times \left[1.0 - \frac{0.7SNI}{SNI+exp(-5.51+22.9SNI)}\right]$$

$(5)$

where $SNI = 1 - SAN/100$, $CLA$, $SIL$, $SAN$ represent the clay fraction (%), silt fraction (%) and sand fraction (%) respectively. $C$ is the organic carbon content (%). The unit of $K$ of US unit (t·acre·h/(100 acre·ft·tonf·in)) can transfer to the SI units (t·ha h/(ha·MJ·mm)) by multiplying by 0.1317.

### 2.3.3. Topographic Factors (LS)

The LS factor in the USLE represent the relationship between the soil erosion and topography (slope length and slope gradient) [24]. Higher slope gradient or longer slope length can accelerate the velocity of raindrops and extent of the time of water flow and then impact the soil surface particles directly and cause the most severe soil erosion [28]. In this paper, we adopted the method [18,30] for the slope in the research region exceeding 20% rather than the algorithms used in the RUSLE limited the slope ≤18% [31]. The equation can be expressed as:

$$L = \left(\frac{\gamma}{22.1}\right)^m \begin{cases} m = 0.2 & \theta \le 0.5° \\ m = 0.3 & 0.5° < \theta \le 1.5° \\ m = 0.4 & 1.5° < \theta \le 3° \\ m = 0.5 & \theta \ge 3° \end{cases}$$

$(6)$

$$S = \begin{cases} 10.8sin\theta + 0.03 & \theta < 9\% \\ 16.8sin\theta - 0.5 & 9\% \le \theta \le 18\% \\ 21.9sin\theta - 0.96 & \theta > 18\% \end{cases}$$

$(7)$

where $\gamma$ is the slope length (m) and $m$ is depending on the slope percent ($\theta$).

### 2.3.4. Soil Conservation Measures (B, T and E)

The dimensionless factors of slope and soil conservation measures were defined as the ratio of soil loss from the unit plot to actual plot with the aimed factor changed but the same sizes of other factors as the unit plot. The Chinese soil loss equation is used to estimate the annual average soil loss based on the from slope cropland considering the different land uses and different soil conservation practices [13]. In this paper, land uses interpreted by the Arial scanning image, Google image and high-resolution image based on the ArcGIS environment. Vegetation coverage (f) was derived from the Landsat MSS and Landsat TM scenes based on the Google Earth Engine (GEE) platform, a cloud-based large-scale computational facility for geospatial data analysis, provided us a rare opportunity to obtain earth observation data more accuracy and quickly and access remote sensing imagery freely. The processes can be divided as the following steps: (i) removing the cloud and shadow pixels according to the pixel data quality of flag information and (ii) adding the band of NDVI for every image according following equation:

$$NDVI = (NIR - RED)/(NIR + RED)$$

$(8)$

where NIR and RED refer to the surface reflectance in the near infrared band (band 4—Landsat 5, 7; band 5—Landsat 8) and red band (band 3—Landsat 5, 7; band 4—Landsat 8) and (iii) combining all

Landsat-derived NDVI and extracting NDVI according to the study destination (study area and study time series). (iii) Calculate the vegetation coverage

$$f = \frac{NDVI - NDVI_{soil}}{NDVI_{max} - NDVI_{soil}}$$

(9)

$NDVI_{soil}$ and $NDVI_{max}$ represent the pure bare soil pixels and pure vegetation pixels respectively.

The B value estimated by integrating the vegetation covers and land uses types according to the USLE manual and the method proposed by Cheng [17] (Table 2).

**Table 2.** B factor values with different land uses and vegetation cover.

| Land uses Type | Vegetation Cover | B |
|---|---|---|
| Forest | 0%–20% | 0.1 |
| | 20%–40% | 0.08 |
| | 40%–60% | 0.06 |
| | 60%–80% | 0.02 |
| | 80%–100% | 0.004 |
| Building | - | 0.9 |
| Water | - | 1 |
| Grassland | 0%–20% | 0.45 |
| | 20%–40% | 0.24 |
| | 40%–60% | 0.15 |
| | 60%–80% | 0.09 |
| | 80%–100% | 0.043 |
| Cropland | - | 0.23 |

E value considering the method used in the Yanhe River basin [32], the formula is expressed as:

$$E = (1 - \frac{S_t}{S} \times \alpha) \times (1 - \frac{S_d}{S} \times \beta)$$

(10)

where $S_t$ is the area of terrace, $S_d$ is the area controlled by the checke-dams, $S$ is totally soil area, $\alpha$, $\beta$ are the sediment reduction coefficient of terrace and dams respectively and their value is defined as 0.764 and 1 according to the previous researches [33,34].

T factor decided by the amount of soil loss reduction by contour tillage under the different slope condition (Table 3).

**Table 3.** T value under the different slope conditions.

| Slope Range | 0° | ≤5° | 5–10° | 10–15° | 15–20° | 20–25° | >25° |
|---|---|---|---|---|---|---|---|
| T value | 1 | 0.1 | 0.221 | 0.305 | 0.575 | 0.705 | 1 |

## 3. Results

### 3.1. Soil Erosion Estimation in the Jiuyuangou Watershed

Figure 2 shows the spatio-temporal distribution and changes of soil erosion in the JYG watershed in 1977, 2004 and 2015. The results show that the soil erosion was 2011 (t/km$^2$·a), 164 (t/km$^2$·a) and 320 (t/km$^2$·a) for 1977, 2004 and 2015, respectively. In this paper, soil erosion intensity classified into the seven grades according to the level of soil erosion severity (Table 4). The corresponding grade of I, II, III, IV, V, VI and VII were no significant, slight, light, moderate, intensive, very intensive and severe soil erosion, separately. The researches demonstrated that the soil erosion in 1977 was mainly dominated by grade II, III, IV and V, which accounted for 28.40%, 17.64%, 22% and 17.11% of the total soil erosion

modulus, respectively. However, in 2004 and 2015, grade I became the mainly soil erosion grade in JYG watershed, which accounted for 88.98% and 69.34% of the total soil erosion modulus, separately.

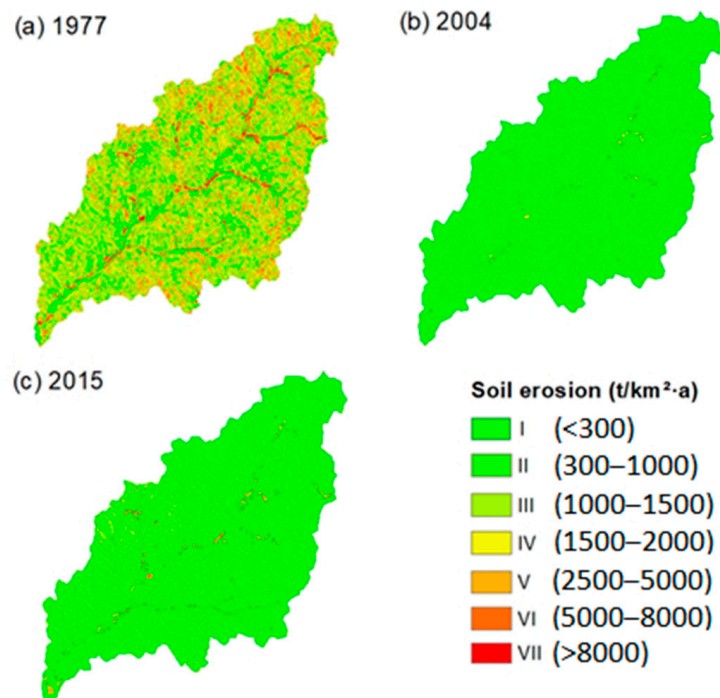

**Figure 2.** Soil erosion distribution in the JYG watershed in (**a**) 1977, (**b**) 2004 and (**c**) 2015.

**Table 4.** Distribution of grades divided by soil erosion intensity from 1977 to 2015 in the JYG watershed.

| Grade | Level | Range (t/km$^2$·a) | 1977 | | 2004 | | 2015 | |
|---|---|---|---|---|---|---|---|---|
| | | | Area (km$^2$) | % | Area (km$^2$) | % | Area (km$^2$) | % |
| I | No significant | <300 | 620.14 | 8.93 | 6184.89 | 88.98 | 4814.10 | 69.34 |
| II | Slight | 300–1000 | 1972.19 | 28.40 | 626.70 | 9.02 | 1858.05 | 26.76 |
| III | Light | 1000–1500 | 1225.09 | 17.64 | 89.92 | 1.29 | 128.80 | 1.86 |
| IV | Moderate | 1500–2500 | 1527.99 | 22.00 | 23.80 | 0.34 | 61.27 | 0.88 |
| V | Intensive | 2500–5000 | 1188.23 | 17.11 | 22.25 | 0.32 | 48.95 | 0.71 |
| VI | Very intensive | 5000–8000 | 263.04 | 3.79 | 2.96 | 0.04 | 21.73 | 0.31 |
| VII | Severe | >8000 | 148.44 | 2.14 | 0.65 | 0.01 | 9.42 | 0.14 |

In this study, we drew an area change histogram plot of different soil erosion intensity grades to illustrate the conversion of soil erosion intensity more visually (Figure 3), we could find that the area of soil erosion that converted from other grades to grade I from 1977 to 2015 (area of flow-in of I) was the largest, which occupied 69.3% of total change area from 1977 to 2015, followed by another grade that converted to grade II (area of flow-in of II), which accounted for 26.83% of the total change area. Specifically, many parts of grade II, III, IV and V were transferred to grade I, which accounted for 35.78%, 18.24%, 20.47% and 10.98% of the total area change, respectively. Noting that the soil erosion area of grade VI and VII just accounted for 3.79% and 2.14% and 0.31% and 0.14% of the total soil erosion in 1977 and 2015, it indicated that the transfer proportion of grade VI and VII to grade I was still large in the whole study period.

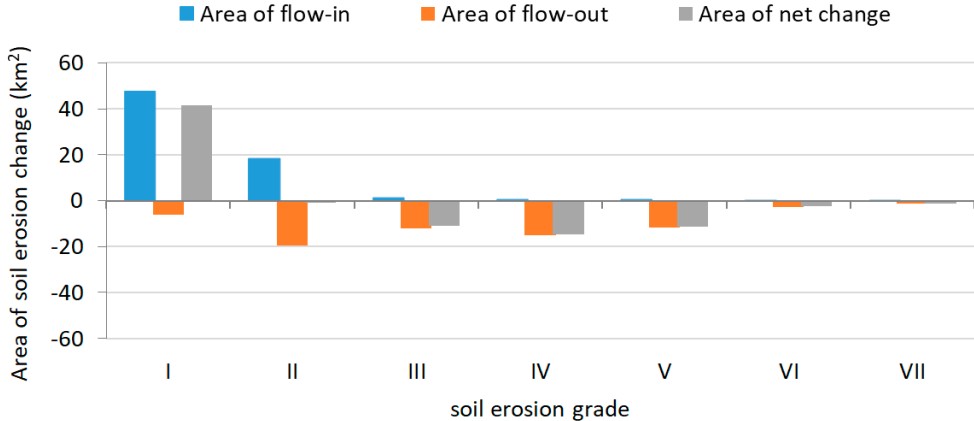

**Figure 3.** Area change in a different grade from 1977 to 2015 (the area of flow-in represents the area of change from 1977 to 2015, the area of flow-out refers to the opposite conversation and the area of net change was the total area change from 1977 to 2015).

### 3.2. Land Use/Cover Changes and Its Effect on Soil Erosion

The spatial distribution of different types of land use on the JYG watershed in 1977, 2004 and 2015 are shown in Figure 4. Evidence shows that the main land uses type were farmland and grassland in the JYG watershed, and it had a great change with a decrease in farmland and increase in grassland from 1977 to 2015. Table 5 depicts the average soil erosion modulus and average vegetation covers corresponding to the different land uses type in 1977, 2004 and 2015. The results show that the rate of change of soil erosion for different land use in a decreased order from high to low as: forest > residential area > road > farmland > grassland. The area of farmland was declining in recent years, which decreased from 47.97% to 16.59% of the total study area. This change played an important role in helping to relieve the soil erosion in the basin, which showed a reduction trend from 1195.89 to 164.81 (t/km$^2$·a). Although the area of forest was small in the basin, its soil erosion reduced significantly, from 1029.95 to 288.01 (t/km$^2$·a) with a 27.96% change rate. The grassland is the main land use type with a moderate and low vegetation cover in the study area. Therefore, the soil erosion of grassland was higher compared with that of forest and farmland. Noting that the road and residential area also presented the large soil erosion in the basin, which was 8838.77 (t/km$^2$·a) and 11,262.8 (t/km$^2$·a) in 1977 respectively.

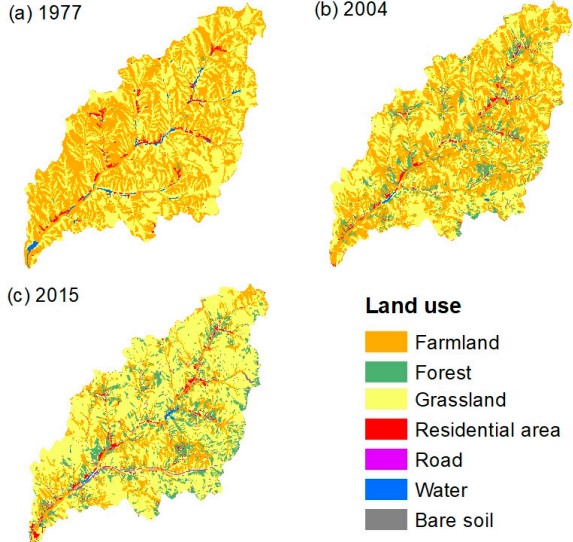

**Figure 4.** Spatial distribution of land uses during 1997–2015 in the study area

**Table 5.** Soil erosion and vegetation cover of different land uses in 1977, 2004 and 2015 in the study area.

| Land Use Types | Area% | | | Vegetation Coverage | | | Soil Erosion Loss (t/km$^2$·a) | | | |
|---|---|---|---|---|---|---|---|---|---|---|
| | 1977 | 2001 | 2015 | 1977 | 2004 | 2015 | 1977 | 2004 | 2015 | Rate of Change (1997–2015) |
| Farmland | 47.97 | 39.87 | 16.59 | 35.57 | 39.16 | 43.76 | 1195.89 | 105.58 | 164.81 | 13.78 |
| Grassland | 49.08 | 47.57 | 64.68 | 37.46 | 37.17 | 47.58 | 2386.75 | 213.35 | 319.52 | 13.39 |
| Forest | 0.24 | 15.60 | 22.46 | 44.26 | 38.54 | 50.33 | 1029.95 | 178.89 | 288.01 | 27.96 |

### 3.3. Topography and its Effect on Soil Erosion

According to the soil erosion classification standard SL_190-2007, slope gradients classified into six grades 0–5°, 5–8°, 8–15°, 15–25°, 25–35° and >35° respectively. Table 6 shows the rates of soil erosion corresponding to different grades of slope. The results imply that the soil erosion showed an increased trend with the slope gradients increased. In contrast to the slope of 0–5° with the lower erosion rate the steep slope area (slope of >25°) showed the highest erosion. For example, the soil erosion of slope of 0–5° in 2015 was 29.45 (t/km$^2$·a), which was far less than that of slope of more than 25° with a soil erosion exceeding 282.27 (t/km$^2$·a). Similarly, the LS factors also had a positive correlation with the slope gradient. For example, the lowest mean LS value was found in the slope of 5–8°, and higher LS value were always occurring in the higher slope.

**Table 6.** Soil erosion of different slope gradients in 1977, 2004 and 2015 in the study area.

| Slope | Area | LS Factor | | Soil Erosion (t/km$^2$·a) | | |
|---|---|---|---|---|---|---|
| | km$^2$ | Mean | std | 1977 | 2004 | 2015 |
| 0–5° | 152.24 | 0.50 | 1.48 | 355.73 | 63.06 | 29.45 |
| 5–8° | 152.74 | 1.20 | 1.82 | 603.68 | 98.08 | 48.38 |
| 8–15° | 501.12 | 2.92 | 2.49 | 904.18 | 146.57 | 73.15 |
| 15–25° | 1441.94 | 6.75 | 3.70 | 1652.46 | 242.89 | 131.99 |
| 25–35° | 2281.13 | 11.25 | 4.84 | 2001.29 | 333.09 | 165.12 |
| >35° | 2436.28 | 15.96 | 6.18 | 2650.71 | 418.21 | 217.15 |

Different slope and soil erosion intensives had a close relationship under the same land uses type in the JYG watershed from 1977 to 2015. Figure 5 shows these relationships and indicates that the soil erosion became more serious with the increase of slope gradients under the same land uses. Especially from 1977 to 2004, soil erosion decreased significantly with the slope increases and since 2015, this trend of increase gradually became slow. Additionally, under different land use patterns, soil erosion also presented the same regularity with the increase of slope, that is, decreased drastically first and then increased slowly or remained stable.

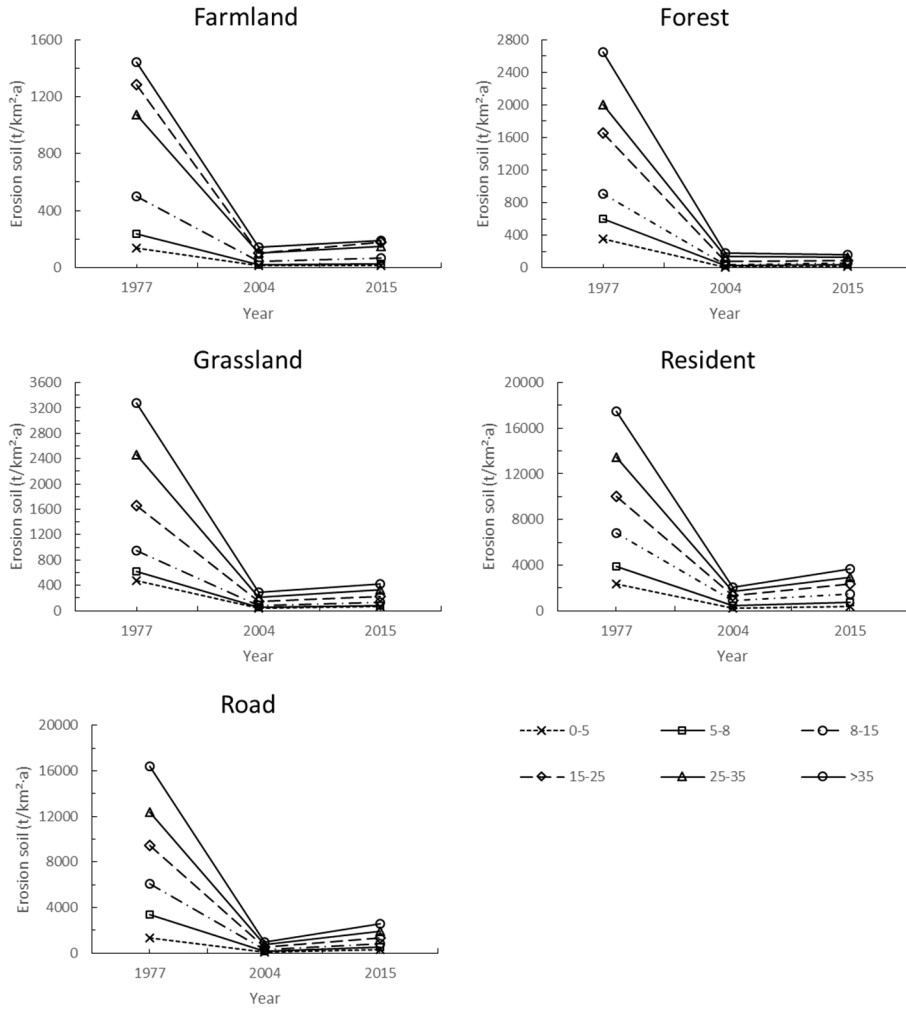

**Figure 5.** Soil erosion of different slope gradients under different land uses in the JYG watershed from 1977 to 2015.

## 4. Discussion

### 4.1. Climate Change

Considering how better to control soil loss in the future, the relationship between soil erosion and rainfall intensity needs to be watched [35]. The average annual soil erosion calculated in this study up to 2011 (t/km²·a) in 1977 was far greater than that in 2004 and 2015 (164 (t/km²·a) and 320 (t/km²·a)). The mean soil erosion of the same land uses increased first then decreased in 1977, 2004 and 2015. Those inclinations at some degrees related to the precipitation variable, which were with the higher rainfall intensities and amounts in 1977. In 1977, the annual average rainfall was 537.3 mm within the study area, and in the flood-season the rainfall lasted 18.81 hours with 129.1 mm and average rainfall intensity up to 0.11 mm per minutes, of which the maximum intensity was up to 1.1 mm per minute and was delayed 8 minutes [36]. In 2004 and 2015, in contrast, there was only 256.9 mm and 334.6 mm average annual rainfall. These statistical data imply that the years with higher rainfall amounts were more inclined to generate severe annual erosion (Table 7).

**Table 7.** Statistical characteristic value of rainfall in the JYG watershed in 1977, 2004 and 2015.

| Year | Maximum (mm) | Minimum (mm) | Average (mm) | Standard Deviation |
|------|--------------|--------------|--------------|--------------------|
| 1977 | 689 | 537.3 | 598.21 | 31.9 |
| 2004 | 334.90 | 201.35 | 256.93 | 21.95 |
| 2015 | 487.84 | 222.8 | 334.6 | 55.64 |

On 26 July 2017, an extreme rainstorm occurred in the JYG watershed and drew a great deal of national attention. According to the statistical data from "Suide Soil and Water Conservation Science Test Station of Upper And Middle Yellow River Bureau, Yellow River Conservancy Commission", the average rainfall within the research area lasted 9 hours with 122.3 mm and maximum stream flow reached 9.5 m$^2$/s in the Delta Trough. The maximum peak water level was 9.5 m$^2$/s and the large section was up to 0.67 m. According to the analysis of 50 years of rainfall data in the rainstorm center of Zizhou, Suide and Mizhi three counties, Liu et al., considered that the daily rainfall and 24 h accumulated rainfall of this heavy rainfall occurred once in 500 years [37]. After a field investigation, the rill developed was noticeable in the steep slope farmland and even produced some small cut ditches. Whereas the forestland converted from farmland in 2000 and the natural restoration grassland was dominated by surface erosion and no obvious rill was produced [38]. This shows that after more than 40 years of efforts by soil and water conservation workers, continuous soil and water conservation measures such as afforestation and conversion of farmland to forests have made the Loess Plateau reach a high degree of governance and good vegetation coverage [37].

### 4.2. Land uses and Vegetation Covers

What is more, the process of soil erosion is not just influenced by rainfall intensity but also links to the variation of land uses [39]. The area of farmland in the JYG basin showed an evidence reduction trend. According to the relationship between the land uses with soil erosion, it indicated that the reduction of farmland mitigated the soil erosion. According to the field investigation, we could confirm that the measures such as the "change of slope farmland into grassland or terrain" and a "limited pledge in slope, which was higher than 25°" were effective to reduce soil erosion over the last ten years. In regards to the soil erosion caused by grassland in 1977, which was much higher than that caused by farmland and forest, we thought that this was due to the vegetation covers for grassland mainly were of low and moderate grade in the catchment. According to the statistics of the vegetation covers in the catchment (Table 5), we found that the vegetation cover in 1977 was 37.5% and mainly had a distribution in the range of medium-low coverage (30%–45%), and then increased to 47% in 2015 because the most of the vegetation coverage change from the original medium-low coverage to medium-high coverage (45%–60%), which mainly benefit from the "Grain For Green Project" (GFGP) implemented by the Chinese central government in 1999. Some effective measures such as planted grassland and afforestation under the close hillsides, which aid the vegetation cover restoration on the Loess Plateau and realized the initial success over the last ten years [40,41].

### 4.3. Slope

Slope is one of the considerable impact factors that affect soil erosion processes [42,43]. Steep farmland refers to the areas of cultivated in the slope up to 25°. According to statistics, about three-fourths of farmland existed as steep slope on the Loess Plateau, which also play an important role in accelerating soil erosion [44]. As in this study, the soil erosion in the JYG watershed shows a serious trend with the increase of slope (Figure 5). In 1977, soil erosion exceeded 2000 (t/km$^2$ a) where the slope was greater than 25 degrees, much larger than where the slope was less than 25 degrees. The soil erosion of the slope that up to 25° change from 4652 to 751.3 (t/km$^2$·a) during 1977 and 2015 (Table 6), and their difference up to six times. Additionally, the soil erosion under the same land uses also presented the same decreased trend from a higher slope to a lower slope (Figure 5). This improvement

is owed to a large number of measures such as limited tillage on steep slope and changes the steep farmland to terrains employed on the Loess Plateau by Chinese government since 1999 [45]. The return of farmland to forests and grasses has obvious erosion reduction effects, but steep-slope cultivation still exists and soil erosion is severe, which is deduced by many experts after field surveys for extreme rainstorm on 26 July 2017 [46]. It can be spectacular that the main reasons of soil erosion representing the severe situation were the existence of the large area of farmland and the grassland (mainly moderate and low vegetation covers) from 1977 to 2015 in the JYG watershed. So, environmental protection measures and soil and water conservation should continue to strengthen and intensify in the fragile ecological area, especially in steep slopes and reduce the area of steep farmland.

## 5. Conclusions

This study evaluated the soil loss changes from 1977 to 2015 in the JYG watershed and analyzed the relationship between the soil erosion and land uses, vegetation cover and slope. The results can be summarized as follows:

The amount of soil erosion in JYG decreased from 1977 to 2015. Specifically, the annual average soil erosion decreased drastically from 2011 (t/km$^2$·a) in 1977 to 164 (t/km$^2$·a) in 2004 and then increased to 320 (t/km$^2$·a) in 2015. Additionally, the areas of soil erosion of grades I (no significant soil erosion) was mainly existed soil erosion form and increased dramatically from 8.93% to 69.34% from 1977 to 2015. Besides, the area of forest only was 0.24 km$^2$, which was smaller than farmland and grassland, but its ability to mitigate soil erosion was higher than both. The area of percentage of grassland accounted for 37% of the total area, and its soil erosion was higher than farmland and forest. Moreover, slope was a key factor that caused a direct effect on soil erosion losses. The soil erosion increased gradually along with slope increases under all kinds of land-use type.

In recent years, sustainable soil and water conservation measures have been effective on the Loess Plateau. However, after the extreme rainstorm on 26 July 2017, many researchers found that the steep slope farmland still exists in a widespread area and its soil erosion is also serious on the Loess Plateau. Additionally, due to the fragility of roads and the existence of steep farmland, the continuous promotion of soil and water conservation measures such as increasing grassland vegetation coverage and reducing the existence of sloping farmland and improve the road quality is still a key step to prevent erosion and reduce soil erosion in the Loess plateau.

**Author Contributions:** Data curation, C.H., X.C. and Y.L.; Formal analysis, Q.Y. and X.C.; Resources, Q.Y.; Supervision, Q.Y.; Validation, Q.Y. and Y.L.; Visualization, C.H.; Writing—original draft, C.H.; Writing—review & editing, C.H., Q.Y., X.C. and Y.L. All authors have read and agreed to the published version of the manuscript.

**Funding:** This study was supported by the Strategic Priority Research Program of Chinese Academy of Sciences, Pan-Third Pole Environment Study for a Green Silk Road (XDA20040202).

**Acknowledgments:** The author is grateful to the editor and anonymous reviewers for spending their valuable time on constructive comments and suggestions that improved the quality of the manuscript considerably and thanks CSIRO Land & Water for their supports.

**Conflicts of Interest:** The authors declare no conflict of interest.

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
