# Peer review of "Assessment of the Soil Erosion Response to Land Use and Slope in the Loess Plateau—A Case Study of Jiuyuangou"

_water, doi:10.3390/w12020529_

Round 1

Reviewer 1 Report

This paper investigates the soil loss changes from 1977 to 2015 in the Jiuyuangou of Loess Plateau, using the Chinese Soil Loss Equation. The scientific contribution of this paper is solid, and it provides interesting content to a broad range of readers. However, considerable revision is required before this manuscript could be published.

Foremost, the language of this paper must be improved. The readability of this paper has been significantly undermined by numerous grammar mistakes. Taking the first sentence in Introduction as an example, I do not think soil erosion could lead to the carbon balance. Moreover, “… in the Loess Plateau mainly for (to) improve and conserve…” and “… researchers have analyses (analyzed)…” should also be corrected. The inconsistency of tense also makes this paper more confusing. Authors should pay more attention to the language of this paper.

Second, authors need to be consistent with the term usage of “LS” (Line 111) and the separate factors of “L” and “S” in equation (1). It is a little confusing in Section 2.3.3.

At last, I recommend authors to rewrite Equation (1), using “✖️” as the delimiter to separate each factor.

To summarize, authors need to improve the language of this manuscript, and the proof-reading from a native speaker is highly recommended for the revision.

Author Response

The language of this paper must be improved. The readability of this paper has been significantly undermined by numerous grammar mistakes. Taking the first sentence in Introduction as an example, I do not think soil erosion could lead to the carbon balance. Moreover, “… in the Loess Plateau mainly for (to) improve and conserve…” and “… researchers have analyses (analyzed)…” should also be corrected. The inconsistency of tense also makes this paper more confusing. Authors should pay more attention to the language of this paper.

Re: We thank the reviewer for this comment. Considering the language problems, we invited persons who good at English writing and native speaker to give us some suggestions and revise it again. The revise sentence all labelled by using “Track Changes” in the manuscript.

Second, authors need to be consistent with the term usage of “LS” (Line 111) and the separate factors of “L” and “S” in equation (1). It is a little confusing in Section 2.3.3. At last, I recommend authors to rewrite Equation (1), using “✖️” as the delimiter to separate each factor.

Re: We already revised in the line 123. L and S refer to the slope length and slope gradient, separately. Those two factors represent the relationship between the soil erosion and topography. According to the comment, we already use “✖️” to separate each factor in Equation (1).

To summarize, authors need to improve the language of this manuscript, and the proof-reading from a native speaker is highly recommended for the revision.

Re: Thanks for your reviews. We already invite some persons to help us revise grammar and tenses for this manuscript.

Additionally, we found the number of figure and table is wrong, so we already revised using “Track Changes” in the manuscript.

Reviewer 2 Report

Soil erosion is a serious environmental problem in many countries around the world. this article presents research over a long period of time that covers the years 1977-2015. The reviewed article shows reliable data and information on changes in the environment that occur during the discussed phenomenon in nature. The work refers to a large number of references, which is very good theoretical preparation of the manuscript. The structure of the article is correct, all arranged in a transparent way for the reader.
Tables, figures, drawings, maps of very good quality, prepared in a clear and easy to interpret manner.
I propose to accept it for publication in its current form.

Author Response

We really appreciate the reviewer for the recognition of this paper.

Round 2

Reviewer 1 Report

Thank you for addressing all my comments.